# Implementing E-Cigarettes as an Alternate Smoking Cessation Tool during Pregnancy: A Process Evaluation at Two UK Sites

**DOI:** 10.3390/ijerph21030291

**Published:** 2024-03-01

**Authors:** Eleanor Lutman-White, Riya Patel, Deborah Lycett, Kelly Hayward, Ruth Sampson, Janani Arulrajah, Maxine Whelan

**Affiliations:** 1Centre for Healthcare and Communities, Coventry University, Coventry CV1 5RW, UK; ad6406@coventry.ac.uk (E.L.-W.); rp526@leicester.ac.uk (R.P.); ab5042@coventry.ac.uk (D.L.); 2Centre for Ethnic Health Research, NIHR Applied Research Collaboration-East Midlands (ARC-EM), University of Leicester, Leicester LE5 4PW, UK; 3Warwickshire Public Health Team, Warwick CV34 4RL, UK; 4Bath and North Somerset Public Health Team, Bristol BA1 1JQ, UK

**Keywords:** implementation, process evaluation, smoking cessation, pregnancy

## Abstract

Smoking during pregnancy increases the risk of adverse maternal and foetal health outcomes, with effective smoking cessation support important. E-cigarette use in the general population has increased rapidly in recent years, with their use viewed as an alternate, additional offer to nicotine-replacement therapy and behavioural support. However, their use in pregnancy has limited investigation. This study aimed to understand how two e-cigarette pilots for pregnant women were delivered and implemented. Referrals to the general stop smoking in pregnancy service, as well as pilot enrolment, engagement and outcomes were recorded. Seven professionals involved in pilot 2 design, setup and/or delivery took part in semi-structured interviews informed by the Consolidated Framework for Implementation Research (CFIR). Transcripts were deductively coded into CFIR. In total, 124 of 296 women accessed at least one visit after being contacted and offered the e-cigarette pilot (Pilot 1: N = 99, Pilot 2: N = 25). In Pilot 2, 13 (of 25) reached 4 weeks, and common reasons for withdrawal by 12 weeks included relapse, loss of contact and no further support wanted. Forty-five (36.3%) validated quits were reported (Pilot 1: 32 of 99 (32.3%); Pilot 2: 13 of 25 (52%)). Facilitators included regular communication and the advisors physically taking e-cigarettes to home visits. Barriers included misalignment between the pilot and the standard treatment offer and availability of the staff resource. Enrolment to both pilots was demonstrated, with greater enrolment in one pilot and notable quit rates among women across both pilots. The perceived role of e-cigarettes for pregnant women varied, and a lack of staff resources explained some challenges. Adaptations may be needed during scale-up, including additional resources and the alignment of the e-cigarette provision to standard treatment.

## 1. Background

An estimated 8 million deaths are attributed to smoking worldwide annually [1,2], with the global smoking prevalence in pregnancy estimated to be at 1.7% [3]. Concerningly, nearly one in ten pregnant women in England were smokers at the time of delivery in 2022–2023 [4]. Smoking in pregnancy has been attributed to several, severe negative foetal health outcomes (e.g., stillbirth, preterm birth, congenital malformation) [5]. Although many women quit after discovering their pregnancy [6], a proportion of women continue to smoke because they are dependent on nicotine and find it difficult to quit [7]. With more than 200 million pregnancies estimated to occur worldwide annually [8], smoking cessation support during pregnancy is a worthwhile endeavour to minimise adverse pregnancy and perinatal outcomes.

In the UK, nicotine replacement therapy (NRT) is widely recommended and routinely offered to pregnant women who are unable to quit smoking unassisted [9,10]. NRT delivers a controlled amount of nicotine without the chemicals found in cigarettes and is often used to reduce the urge to smoke. In pregnancy, women metabolise nicotine at a faster rate and so this must be factored in to smoking cessation support. A survey in England found that 86% of smoking cessation services offered combination NRT (two forms of NRT together) in pregnancy to support smoking cessation [11]. The use of NRT is also in alignment with the UK’s “Saving Babies Lives” bundle, which targets reducing perinatal mortality [12]. Evidence of treatment efficacy for NRT in this population is, however, weak. When pregnant women are prescribed NRT in trials, only 7–29% reported finishing prescribed NRT courses [13]. The level of adherence to NRT has been reported as being lowest among pregnant women [14]. This poor adherence may be partially due to concerns about using NRT [15] and increased nicotine metabolism during pregnancy [16]. A systematic review of seven randomised controlled trials (RCTs) and 23 non-RCTs declared unclear evidence as to whether the maternal use of NRT during pregnancy is harmful to the foetus (e.g., neonatal death, preterm birth) [17]. Aligning with up-to-date clinical guidance, the expert opinion is that using NRT in pregnancy is much safer than continuing to smoke [13]. Wider literature suggests that behavioural interventions, such as counselling, feedback and incentives, increase smoking cessation rates in pregnancy [18]. Yet few of those formally tested are proven effective [19]. Another alternate, additional offer to NRT and behavioural support has been the rapid rise in e-cigarette use in people who smoke and are not pregnant [20].

In non-pregnant people, e-cigarette use has been shown, in a randomised trial, to be more effective than traditional NRT for smoking cessation [21]. An increasing prevalence of e-cigarette use has been noted during pregnancy [16,22], with the evidence exploring the harms and benefits of e-cigarette use during pregnancy also increasing. A randomised controlled trial of 1,140 pregnant women concluded that the safety profile of e-cigarettes and NRT is similar; however, a low birthweight (<2500 g) was less frequent in the e-cigarette arm (14.8% versus 9.6%; RR = 0.65, 95%CI: 0.47–0.90, *p* = 0.01) than with NRT [23]. The most recent systematic review of e-cigarette use in pregnancy, published in 2023, suggested that 1.2% to 4.8% of pregnant women used e-cigarettes, and e-cigarettes were widely perceived to be safer than smoking [24]. When pregnant women have been surveyed about their reasons for e-cigarette use during pregnancy, the most common reasons are related to perceptions that they are a less harmful option (compared to cigarettes) and to help with smoking cessation [25]. However, there is currently no clinical knowledge around the efficacy and safety of e-cigarettes in pregnancy [26]. With e-cigarette use considered less toxic than tobacco smoking and a good harm reduction alternative to conventional smoking [27], studies investigating their use as a smoking cessation tool are warranted in pregnant women. Country-specific guidance and practice on offering e-cigarettes in pregnancy for smoking cessation can vary widely. Evaluating the implementation of pilot interventions is important to help guide future practice. This study evaluated the implementation of two UK-based pilots and explored the views of stakeholders about implementation and scalability. Pregnant women’s experiences of using e-cigarettes during pregnancy from the two pilots are reported elsewhere (Lutman-White et al., under review).

## 2. Methods

### 2.1. Study Design

An evaluation of routinely collected data on pilot enrolment, retention and effectiveness with a nested qualitative study with professionals who were involved in pilot design, setup and/or delivery was performed. The nested qualitative study was written up in accordance with the Consolidated Criteria for Reporting Qualitative Research checklist [28].

### 2.2. Context

Pilot 1 and 2 were delivered at two different UK sites. Regardless of the site, women were referred to a specialist stop smoking in pregnancy service (Figure 1). Both stop smoking in pregnancy services set out to provide an additional offer to the standard offer of NRT and behavioural support to pregnant women. Both pilots aligned with recommended practice where pregnant women should be offered two forms of nicotine-replacement therapies during a structured quit attempt (typically patches to match nicotine levels the body is used to and another form to cope with the physical cravings, in this instance, through e-cigarettes). Pregnant women were contacted by the stop smoking in pregnancy service team and, if eligible and appropriate, offered a free e-cigarette.

In pilot 1, pregnant women were supplied with a single use, disposable e-cigarette (provides 320 × 2 s puffs; roughly equates to 30–35 cigarettes; approx. 2–4 days), alongside NRT if they wished to receive both. Women were supplied with up to two e-cigarette devices a week, depending on smoking habits and verified by CO readings (to deduce smoking habits objectively by detecting carbon monoxide in exhaled breath). Pregnant women were initially seen weekly at home, with appointments each lasting 30–60 min. During these visits, behavioural support was offered, including information on how to deal with cravings, distraction techniques and what happens to your body when you quit, as well as relapse prevention strategies. After week 4, if relapse had occurred, then visits remained weekly (otherwise fortnightly). Pregnant women also had access to follow-up calls and texts with the advisor as and, when needed, in between appointments. This form of ad-hoc support acted as an important motivator and was used for trust-building. This pilot ran from October 2018 to September 2019 and was delivered by multiple advisors in the Health in Pregnancy Team.

In pilot 2, pregnant women were supplied with a refillable and rechargeable pen-style e-cigarette. They received a compulsory, in-person home visit from a stop smoking in pregnancy adviser at 1, 4, 8 and 12 weeks, in addition to receiving regular phone/text support. The content and intention of the home visits and telephone/text mirrored the support described in pilot 1. The 1-week and 8-week visits were additional to the standard offer of NRT and behavioural support. One stop smoking in pregnancy advisor delivered pilot 2, which ran from July 2022 to May 2023 (with a 3-month suspension due to insufficient resources).

### 2.3. Participant Selection: Sampling

For the main evaluation, participants were pregnant women aged 18 years or older who were referred to the existing stop smoking in pregnancy service in two UK hospital trusts. Information about the pilots was offered at the time of booking, and therefore, they were aware of the e-cigarette offer before their first appointment, during which written consent to participate in the pilot was obtained.

For the interviews, eligible participants were intervention-providers involved in the design, setup (e.g., commissioning) and/or delivery of pilot 2. This included local authority staff, stop smoking in pregnancy advisors and NHS Trust service management staff. Staff were invited to interviews during monthly project meetings for pilot 2. Individuals who were interested in participating in the research provided consent via Qualtrics. These participants disclosed contact details, as well as their age, job title and role in the pilot (e.g., design, setup and/or delivery). After consent, all participants were contacted by the research team to arrange a convenient day and time to conduct the interview.

### 2.4. Setting

Each pilot was delivered face-to-face by specialist stop smoking services for pregnant women in the women’s homes, supplemented with regular phone/text support. An implementation strategy employed by both pilots was the provision of behavioural support alongside the e-cigarette intervention.

Interviews with intervention-providers were conducted either online via Microsoft Teams or by telephone, and the participants were offered evening and weekend slots as necessary. Interviews took place several months after pilot initiation to capture substantial experience.

### 2.5. Data Collection

Data were routinely collected by the stop smoking in pregnancy service teams and were related to stop smoking in pregnancy service referrals plus pilot enrolment, retention and outcomes (i.e., number of validated quits) during the pilot. Given that home visits were undertaken in these pilots, the majority of quits were CO verified (otherwise self-reported). In pilot 1, engagement data related to women engaging with at least one visit. In pilot 2, data were collected about engagement at 4, 8 and 12 weeks.

The semi-structured interview schedule was based on the domains in the Consolidated Framework for Implementation Research (CFIR) [29]. CFIR has five domains (intervention characteristics, outer setting, inner setting, characteristics of individuals and process of implementation) and associated constructs. The inner setting included the network of entities that connected the delivery of stop smoking services, such as NHS services, Public Health, and the local authority. The delivery of the schedule was tailored to the participants’ involvement in pilot 2 design, setup and/or delivery. Interviews were transcribed either using the inbuilt transcription function within Microsoft Teams or manually for telephone interviews, with all identifiable information deleted. Transcripts were not returned to participants. The research team checked transcripts for accuracy, and stakeholders were able to check the final selection of quotes used.

### 2.6. Research Team and Reflexivity

One experienced, female qualitative researcher (ELW) with an interest in health inequalities conducted the interviews with staff. The interviews were described to participants as a forum to reflect openly on their experience so that changes could be made to future service delivery.

### 2.7. Data Analysis

Referrals to the general stop smoking in pregnancy service, as well as pilot enrolment, engagement and outcomes, were recorded at both sites and descriptively reported.

Interview data were coded deductively by one researcher (ELW) into the CFIR. A second researcher (MW) reviewed the quotes coded to each of the CFIR domains to offer alternative interpretations and support reflexivity. Microsoft Excel was used to code and organise the interview data. Interview findings were structured based on the CFIR domains. Quotes were not presented with participants’ professional roles to maintain participant anonymity. Verbatim quotes have been modified to aid readability (i.e., removing word repetitions and hesitations). Interview participants often used the term “vape” rather than e-cigarette and we have left the references to “vape” in the quotes.

## 3. Results

### 3.1. Quantitative Findings

The flow of participants through the two pilots is illustrated in Figure 2. In relation to pilot 1 and pilot 2 sites, 160 and 182 women, respectively, were referred to the stop smoking in pregnancy service in the specified timespans. In pilot 1, 99 women accessed at least one visit. Pilot 1 did not differentiate the proportion who used e-cigarettes alone compared to e-cigarettes plus NRT. In pilot 2, 25 accepted the pilot offer with the most common reason for not accessing the pilot relating to already quitting (N = 35). Pilot 2 retention was reported as follows: 13 (of 25) women reached 4 weeks, 8 (of 13) women reached 8 weeks and 7 (of 8) women reached 12 weeks. Reasons for withdrawal (total N = 18) from pilot 2 included relapse, loss of contact and no further support wanted. In pilot 2, most women selected e-cigarettes exclusively. A total of 45 validated quits were reported (pilot 1: 32 of 99 (32.3%); pilot 2: 13 of 25 (52%)).

### 3.2. Qualitative Findings

Seven professionals relevant to the pilot 2 design, setup and/or delivery participated in an interview. The mean age of interviewees was 44 years with roles and responsibilities in a local authority/NHS setting, including commissioners, public health practitioners, stop smoking in pregnancy service advisors and managers of the stop smoking in pregnancy service. Interviews lasted 45 min on average (range 15–76 min). The results are presented in line with the five CFIR domains and associated constructs (Box 1).

Box 1CFIR domains and associated constructs (^a^ indicates domains and constructs identified in this study).
Domain 1: Intervention characteristics
*This domain incorporates key attributes of interventions that can influence the success of implementation.*
Innovation source ^a^Evidence strength and quality ^a^Relative advantageAdaptabilityTrialabilityComplexity ^a^Design quality and packagingCost
2.Domain 2: Outer setting
*This domain captures macro-level factors, such as policy, economic factors and the social context that are external to the inner setting and which influence implementation.*
Needs and resources of those served by the organisation ^a^CosmopolitanismPeer pressureExternal policy and incentives
3.Domain 3: Inner setting
*This domain concerns the setting in which the intervention is implemented. For our analysis, we have included in the inner setting the network of entities that are connected with the delivery of stop smoking services, such as NHS services, Public Health and WCC as the commissioner of stop smoking services.*
Structural characteristicsNetworks and communication ^a^CultureImplementation climate ^a^Readiness for implementation ^a^
4.Domain 4: Characteristics of individuals
*Organisations are made up of individuals, and the actions and behaviours of these individuals can influence implementation. This domain reports on the individuals involved with the intervention and/or the implementation process.*
Knowledge and beliefs about the intervention ^a^Self-efficacy ^a^Individual stage of changeOther personal attributes
5.Domain 5: Process
*This domain incorporates aspects of the implementation process that contribute to successful implementation.*
Planning ^a^Engaging ^a^Executing ^a^Reflecting and evaluating ^a^


Domain 1: Intervention Characteristics

A combination of internal determinants and external factors influenced the development of the e-cigarette in pregnancy pilot. The team drew on experiences of other local authorities (including language used) and in particular built on pilot 1: “*so we looked at in terms of what [Pilot 1 local authority] had done because obviously we drew a lot of information from what [Pilot 1 local authority] had done and obviously we used a lot of their, cojoined their research paperwork on that as well.”* (intervention provider 6).

One participant noted how talking to other local authorities was helpful in terms of securing the procurement and supply of e-cigarettes: “*so I suppose actually for the procurement side of how other areas were supplying vapes or how they were doing it? Were they doing it on a voucher system? Were they giving them directly, you know. So, there was lots of different ways that different areas were doing these*” (intervention provider 6).

Intervention providers also demonstrated an understanding of evidence supporting the reduced harm from using e-cigarettes compared to combustible cigarettes: “*we know that if someone’s vaping, there’s no carbon monoxide in the [pregnant] lady’s blood, so the baby’s protected, you know, it’s like she’s a non-smoker so in terms of using it for our specific purpose it’s fine*” (intervention provider 1).

This was combined with an awareness that e-cigarette use is not risk-free: “*you just think okay well at least babies are more protected by the [pregnant] woman using a vape then surely that’s the better option, but not the best option, but it’s a better option*” (intervention provider 6). There was less discussion in relation to the evidence around e-cigarettes as a smoking cessation tool in pregnancy.

Interviewees perceived the pilot as a clear departure from existing practices, identifying several differences compared to the standard treatment programme (i.e., NRT). Pilot 2 involved home visits, more visits than as standard and additional paperwork. These aspects increased the time needed to deliver the pilot: “*just trying to get people to undertake the [additional] paperwork and I think one of the things that I’d heard was this particular group of women were interested, but then when you say well it will mean this many home contacts they kind of are like, no, don’t want you come into the home*” (intervention provider 7).

E-cigarettes were also perceived as more complex in terms of options available: “*there was differences in terms of the vape devices, whether it’s like throat to lung or mouth to lung in the style of how they use them, they’re different devices, so it’s all a bit complicated*” (intervention provider 1).

There was widespread understanding about how e-cigarettes are not always perceived as quitting tools by pregnant women, rather they are seen as something that can be used longer-term. For staff involved in Pilot 2, this misunderstanding presented additional challenges:

“*it’s more challenging as an advisor because all the time I’m trying to convince them that this is a temporary measure, we don’t want you then to be addicted to a vape …… it’s very difficult also in terms of behavioural support, I would say that that is much harder to [provide behavioural support], because of their perception of a vape, it’s like a straight swap [for cigarettes]. So where do you put in the behavioural support?*” (intervention provider 2).

Implementing the free e-cigarette offer was perceived as a cost-effective alternate offer: “*There was talk of incentives, but the cost of incentives was going to be, was potentially you know really exceed any budget we had” (intervention provider 6). Some interviewees also raised views about how e-cigarettes and NRT were comparable in terms of product costs: “with vapes versus the NRT products, you know the cost so vapes are cheaper, so that was a factor as well*” (intervention provider 6).

Domain 2: Outer Setting

Interviewees demonstrated an accurate understanding of the needs and resources of those served by the organisation. They were acutely aware of the socioeconomic deprivation, higher smoking rates and health inequalities that exist in the geographical area targeted by Pilot 2. Interviewees were also able to describe how this influenced pilot development and the impact on these inequalities if the pilot was successful: “*We’re now getting to the stage where it’s how do we engage with those women who are entrenched in this [smoking], that are really difficult and we’re gonna have to work really hard to make an impact on just a very small percentage. And I think that by offering the vape it might be a lot of work, but you’re probably gonna make an impact*” (intervention provider 1). Interviewees also described how the pilot was particularly needed in one area of the county compared to another area, with the needs of pregnant women influencing pilot implementation:

“*How do we help this group of, this additional 10% of women that you know, because I think I believe we’re down to about 3% [who smoke] in [area A] and we’re down to about 8% [in area B] which is or certainly lower than the national average just about in [area B] but in [area C] we’re a good 4 or 5% still up, although it’s reducing.*” (intervention provider 5).

Intervention providers identified a range of perceived barriers to participation in the pilot, including the influence of family members, previous negative experiences with professionals and digital poverty (e.g., not having phone credit to reply to text messages sent by the stop smoking advisors). The free e-cigarette was the only perceived facilitator of participation reported by the pilot providers, and this was connected to socioeconomic deprivation in the target population.

Domain 3: Inner Setting

Interviewees raised a perceived lack of consistency and frustration across services about perceptions and the acceptance of e-cigarettes. One interviewee identified the incompatibility between offering e-cigarettes to pregnant women and the lack of facilities available for e-cigarette users at hospital sites: “*one of the mixed messaging problems is around whether we are vape friendly [hospital] sites or not, you know to start with in terms of there’s no designated areas for people to go and vape*” (intervention provider 1). There were also differences between the way that the adult stop smoking service and the stop smoking in pregnancy service in the area treat e-cigarette users: “*with the general service as well, I did ask for clarification, but my understanding is that they’re treating people that are vaping as well as smoking at the hospital … that’s a mixed message, you know, within our [stop smoking in pregnancy] service, we’re saying no, vapers are non-smokers, we’re not treating them*” (intervention provider 1).

There were limitations in terms of readiness for implementation. Staff delivering the pilot identified their lack of knowledge about e-cigarettes prior to the pilot: “*we were novices, you know, so we didn’t really know what the products were or how it all worked*” (intervention provider 1). They accessed knowledge and information from the e-cigarette supplier, from visiting an e-cigarette shop and from guidance produced by the National Centre for Smoking Cessation Training. However, the training was not extensive: “*I wouldn’t say there was like a robust training package*” (intervention provider 1) and there was limited time: “*the time factor as well for us, we just got on with it because there was, I don’t know if we’d have had time to go off doing lots of training anyway*” (intervention provider 1).

There were considerable challenges in relation to the availability of resources to deliver the pilot. It was planned that the pilot would be delivered within the existing capacity of the stop smoking in pregnancy service team. However, a major issue, discussed by all interviewees, was low staff numbers (due to sickness), which became an issue during the pilot and continued for longer than expected. Resources were moved from the general adult population stop smoking service to accommodate delivery of the stop smoking in pregnancy service: “*[the general stop smoking service] then had to come in and support pregnant women, pick up some of the caseload because we wanted the pilot to continue…So it meant that [general stop smoking service] could support those that didn’t want vapes and that [stop smoking in pregnancy advisor] could focus on the vapes*” (intervention provider 4). The team was reduced to one adviser at one stage, so the manager paused the pilot for three months: “s*o we had to say, look, we’re a commissioned service we can’t provide the service right now, so we can’t justify continuing with the vape pilot project when we can’t even provide the service, you know, the basic service to our women*” (intervention provider 1). Whilst the staffing issues were unexpected, interviewees reflected on how things could have been done differently to manage staff resources: *“However, we weren’t aware of, I suppose, or I wasn’t aware of the staffing challenges that might [arise], that’s probably a risk that needed to be flagged from the start”* (intervention provider 4). Some interviewees also discussed the possibility of having more staff able to deliver the pilot in the future, rather than a reliance on one person (similar to pilot 1, which had three Health in Pregnancy Advisors delivering the pilot): “*the other staff in the other areas (of the county) do need to know what to do in terms of the vape pilot, what they’re providing, all the information etcetera, because if that one member of staff in [area of county] goes off sick or is away for any long period, then you’re left with no one else who can do it [deliver the pilot]*” (intervention provider 7).

In addition to staffing issues, there were challenges in relation to the costs allocated to the pilot. It was noted that initially there were no costs directly allocated to the pilot. The team identified a solution that involved absorbing project management costs to help fund the pilot and using underspend from other budgets to fund e-cigarette procurement.

Regular meetings took place with a core group of people involved in intervention commissioning, pilot delivery or pilot evaluation and were identified as important to the successful commencement of the pilot: *“I just think that working together as a partnership and the commitment everyone gave to attending those meetings [involving core group of people involved in intervention commissioning, pilot delivery or pilot evaluation] …from the trusts, from local authority, I think has really been the success to it”* (intervention provider 4). The meetings continued throughout and were noted as important to managing the ongoing progress of the pilot: “*I think what was really good was the regular monitoring meetings were in place. They were a lot more frequently to start with, obviously they’ve reduced. They were still frequent… and that definitely helped mitigate and manage some of those risks (referring to staffing issues)*” (intervention provider 4).

Domain 4: Characteristics of Individuals

Compared with the delivery of NRT to pregnant women, interviewees reported that they were much less familiar with e-cigarettes, they did not feel as comfortable with them and lacked confidence in them as a product. Some participants identified differences in the regulatory frameworks governing NRT and e-cigarettes, and this contributed to feelings of unfamiliarity. People that were primarily involved in delivering the e-cigarette pilot noted that, over time, they had now accepted e-cigarettes as part of smoking cessation and they had become more comfortable with them *“I didn’t know anything about vapes and I feel a lot, you know, a lot more comfortable with them*” (intervention provider 2).

Some interviewees shared their enthusiasm for being able to offer pregnant women e-cigarettes and give them a choice: “I think I was I was excited by it. I did want to be able to give vapes to women, you know, I thought that is an important thing to be able to offer” (intervention provider 1). However, interviewees held reservations about long-term use: “*I’m really mindful that, you know, it’s not something that I want to recommend to women ever for long-term use*” (intervention provider 6). Interviewees also commented on colleagues’ perceptions in relation to e-cigarettes. They reported that some colleagues were suspicious of e-cigarettes, they held beliefs about e-cigarette use that were not supported by the evidence base and that e-cigarette use could evoke strong opinions.

Domain 5: Process

Some interviewees discussed the planning that happened before pilot implementation. This included taking steps to ensure that e-cigarettes were not given to pregnant women aged younger than 18 years: “*are we required to ask for ID [to check pregnant women are over 18]? I think that was part of the pilot design in the end was that they [pregnant women] had been booked at [name of hospital] so at least we knew how old they were*” (intervention provider 1). Another interviewee discussed the processes for pregnant women to access e-cigarettes: “*because there’s a lot of discussion around what types of vapes, how we were actually gonna give it out because, you know, some people are giving out vouchers you know physically, so actually the online platform seemed to be the best way for us to do the pilot because it wasn’t, we weren’t rolling out, we were trialling the pilot*” (intervention provider 4). Whilst there was evidence of pre-implementation planning, it was not considered extensive: “*there was sort of choices to be made, but then we didn’t know what choices had been made and it was like the deadline was on top of us before we knew what it was gonna look like*” (implementation provider 1).

One main deviation from the pre-implementation pilot planning stage discussed by most interviewees was the method taken to access the e-cigarettes. Initially, pregnant women in the pilot were given a code to enter on the supplier website, and the e-cigarette would be delivered to them. However, after one pilot participant shared the code with someone outside of the pilot, the adviser began inputting the codes to place the order on their behalf. Later, the adviser began carrying stock so that the pregnant women could receive their e-cigarette and liquids immediately to increase pilot engagement:

“*when you’re going into someone’s house you’ve really got to think about what are you doing for them, why are they letting you in their house? What’s the benefit to them to engage with you? And so if it’s because [stop smoking in pregnancy adviser] needs to come in and get you to sign 4 consent forms or is it because [stop smoking in pregnancy adviser]’s gonna walk through the door and yes, you gotta sign 4 consent forms but also, you’re gonna get your vape as well. So I think that that helps and you know it’s just it means that there’s more reason for them to engage with the adviser if the adviser is actually giving them the physical products.*” (intervention provider 1).

Interviewees reported on the extensive work that had been undertaken to engage key stakeholders and opinion leaders at different stages in pilot planning (including the Director of Public Health, council elected members, senior leaders at two hospital trusts and locality partnership boards). They felt that they had clearly presented the evidence through conducting a systematic review of the existing evidence base [24], located the pilot in a wider programme of work on tobacco control and prepared a position statement on e-cigarettes: “*the evidence was there to support it [the pilot], and the reason we pulled that [the position statement] together partly was because of some of the feedback, some of the questions we were getting, we need to have our house in order, but actually there hasn’t been too many, more support of it really, not too much of any kind of queries*” (intervention provider 4). Interviewees also noted how including key stakeholders in the regular meetings (reported earlier) was important.

Intervention providers reflected on and evaluated the progress and success of the pilot. There were mixed views about judging pilot success. Enrolment in the pilot was not as high as anticipated: “*take up of the project or the product hadn’t been as high as you kind of thought it could be*” (intervention provider 7). Interviewees were unsure of whether the pilot had reduced the quit rate, and it was thought that some pilot participants might not be fully motivated to quit but had engaged for the free e-cigarette: “*I know that people maybe their hearts not really in it, you know, that they say they want to quit and you think, OK, cause people always surprise you, but the downside is, well, I have the freebie [e-cigarette] and I’ll tell you what you want to hear, but actually we lose them at 4 weeks and they’ve not quit*” (intervention provider 2). Interviewees felt that the pilot was a success because the stop smoking in pregnancy service team, through the pilot, engaged with people they would not usually engage with, thereby justifying the figures recruited: “*we are reaching the people that wouldn’t normally engage. So, I’d say it’s not that it’s in great numbers, but I wouldn’t expect it to be in great numbers*” (intervention provider 2).

Interviewees shared what amendments might need to be made if the pilot were to continue. A key point was the need for sufficient staff trained in offering e-cigarettes. One participant queried about the role of disposable e-cigarettes, which might be preferable. Interviewees felt that offering e-cigarettes in the future would need to optimally align with the standard treatment offer and the existing recording system:

“*we have a vape and they can use it alongside of their NRT or as a stand-alone… so if it works alongside our standard treatment programme, I think that’s what would need to happen for it to fit with our database, our recording system how we work, how we frame it and ‘sell’ it*” (intervention provider 2).

Reducing the number of home visits to align with the standard treatment programme raised concerns for interviewees around ensuring access to e-cigarettes: “*we don’t carry stock of NRT, why would we then carry stock of a vape? So, I guess we’d have to really think about then how it was delivered*” (intervention provider 2).

## 4. Discussion

### 4.1. Summary of Findings

A total of 124 pregnant women accepted either of the two e-cigarette pilots. Key reasons for non-acceptance of the pilot were already quitting, the use of NRT or behavioural support instead. Where data were available (for 25 of 124 women), participant retention in the pilot declined steeply across the 12 weeks. Thirteen (52%) women and 32 (32.3%) women had quit from smoking (validated by CO reading) in the two pilots, respectively.

The development of the e-cigarette pilot involved a combination of internal determinants and external influences, which included drawing on other local authority experience. There was also awareness of the evidence base supporting the reduced harm from using e-cigarettes compared to combustible cigarettes. The need for an e-cigarette pilot was also demonstrated through the intervention providers’ in-depth understanding of the needs and resources of those served by the general stop smoking in pregnancy service. The e-cigarette pilot was noted as differing from the standard treatment, with the pilot including more visits and at home, additional paperwork and delivery of the product. E-cigarettes were perceived as more complex (than NRT) and thought to be viewed by pregnant women as for longer-term use (rather than as short-term quitting tools). These differences potentially contributed to feelings of unfamiliarity and uncertainty in the pilot providers, which decreased as time went on. There were barriers and facilitators to pilot implementation arising from the inner setting. Barriers were related to the implementation climate and the lack of consistency across services in relation to perceptions and the acceptance of e-cigarettes. The availability of resources was also an issue at various points during the development and implementation of the pilot arising particularly from staffing issues. Regular meetings with a core group of people involved in intervention commissioning, pilot delivery or pilot evaluation had a positive effect on implementation. Extensive work was undertaken to engage key stakeholders and opinion leaders at different stages, and this positively contributed to successful pilot development. There were mixed views about the progress and success of the pilot. Whilst enrolment of the pilot was not as high as expected, the team felt that they had engaged with people with whom they would not usually engage. Differences between the pilot and standard treatment were cited as barriers to engagement, as were family member influence and previous negative experiences with professionals. During the pilot, changes in delivery occurred where advisers began taking e-cigarettes to initial home visits to help engagement. Looking to the future, pilot providers wanted sufficient resources to be able to deliver an offering that included e-cigarettes with some flexibility built into the team capacity to address unexpected staffing issues that might arise as it did in this pilot. They also wanted to deliver e-cigarettes in line with the standard treatment programme rather than having a separate, misaligned offer in the service.

### 4.2. Strengths and Limitations

There is a paucity of qualitative research examining the views and experiences of professionals implementing e-cigarettes as a smoking cessation tool in pregnancy, and this study offers an important contribution to the evidence base. A particular strength of this study is its mixed-methods design, which has enabled triangulation of the interviews with professionals with the routinely collected quantitative data. Our qualitative findings provide insights into wider considerations of implementation, including how the intervention was developed, the process of implementation and how the intervention interacts with the context in which it is implemented [30]. This aspect of the evaluation was robustly facilitated by the CFIR framework; a comprehensive consideration of constructs is likely to influence implementation. Interviews were conducted with seven pilot providers in one location and therefore may not be generalisable to other localities where e-cigarette pilots have been implemented. However, a strength was that the professionals involved in setup through to delivery were interviewed. Quantitative data only focused on the pilot period, so we cannot comment on quit rates beyond that. Recruitment to pilot 1 was impacted by the suspension of the pilot, which coincided with “Stoptober”, which is a key month for increasing quit attempts [31]. Pilot 1 engagement data were limited to whether women had attended at least one visit based on routine data collection. The pilots managed to engage specific areas of their respective counties (regions) with higher smoking rates, socioeconomic deprivation and health inequalities, and therefore, the results may be generalised to pregnant women sharing similar demographics.

### 4.3. Consistency with the Literature

Just over half of pregnant women enrolled in pilot 2 had quit smoking by 12 weeks (CO verified), whilst a lower quit rate (32.3%) was recorded in pilot 1. The quit rate for pilot 2 (52%) is similar to that reported for pregnant women by the NHS in their statistics on NHS Stop Smoking Services in England (53.6%), although the NHS figure relates to quits at the 4-week follow up and only 13.6% of those quits were CO-verified [32]. This is a key finding and a strength of the pilot 2 work in particular, recording verified quits at 12 weeks. From the review that we conducted [24], we know that the efficacy of e-cigarettes as a tool for stopping smoking during pregnancy remains unclear. Previous research has reported the difficulties of stop smoking in pregnancy services experience in sustaining engagement with pregnant women over time [33]. We found similar findings with only 28% of the women engaging with the whole pilot. The quit rates that were seen in our pilots were similar to those of Chiang and colleagues (2019), who reported the use of e-cigarettes in pregnancy declining at 1 month following initial uptake at baseline [34].

There was a reported need to explore an e-cigarette pilot in these two UK sites. From a development perspective, this study found that professionals reported being aware of the reduced harm offered by e-cigarettes. This was in part because they conducted a systematic review to explore the evidence based around vaping in pregnancy [24]. This was a key process underpinning the final decision to commission pilot 2. The importance of alleviating safety concerns, associated risks and sensitive views with using e-cigarettes among healthcare professionals has been identified as crucial within and beyond our findings. Farrimond and Abraham (2018) conducted 25 interviews with cessation service staff (advisors, commissioners and managers) in the UK [35]. They reported staff concerns about the negative health effects of e-cigarettes, as well as their safety and the lack of licenced products. Concerns also extended to habits and long-term use, which was discussed in our study where professionals thought pregnant women were perceiving the e-cigarettes to be available for longer-term use. This is a significant factor that can pose a barrier if not addressed before the wide-scale roll out of e-cigarettes across stop smoking services in pregnancy and beyond. Professionals also commented on the need to order and deliver vapes and liquids to service users; Farrimond and Abraham (2018) discussed practical barriers such as the unavailability of e-cigarettes based on a prescription [35]. Our findings indicated inconsistency across healthcare services about the perceptions and acceptance of e-cigarettes and misconceptions held by NHS staff about e-cigarettes. Previous research investigating healthcare professionals’ beliefs and attitudes towards e-cigarettes during pregnancy and postpartum indicated a lack of knowledge amongst healthcare professionals, a fear of vaping and, in some instances, the equivalence of e-cigarette users with smokers [36]. This can also extend into general cessation services with professionals not fully convinced about the public health shift toward e-cigarettes [35]. Professionals noted that the training on e-cigarettes that they received was not extensive, reflecting the findings of Hunter et al. (2021) that there was a lack of training available in the area [36]. Explorations in relation to the CFIR construct of readiness for implementation declared scarce resources at various points during the development and implementation of the pilot. Previous research exploring implementation suggested that the quantity of resources available for implementation and ongoing delivery are positively associated with effective implementation [37,38,39]. Professionals in our study identified several barriers to pregnant women engaging in the e-cigarette pilot, including the home visits, the influence of family members and previous negative experiences with professionals. Previous studies have also identified a lack of support from family members and partners as a barrier to smoking cessation [40,41], although these studies did not specifically focus on e-cigarette use for smoking cessation. Changes to improve the burden of delivery could include reducing the number of visits, encouraging family members to engage in the service in parallel and exploring other options for carbon monoxide monitoring (such as a midwife or self-testing). With only small numbers of staff involved in pilot delivery, the training of staff would also be an important factor. This would help to facilitate the emergent e-cigarette friendliness reported by staff working in a general smoking cessation context [35]. There is also a clear need for parity of treatment across stop smoking in pregnancy services and general adult stop smoking services around e-cigarettes and ensuring hospital sites are “vape-friendly” if this is to be scaled up and rolled out widely across the country. For example, a Royal College of Physican’s Report in 2018 suggested the permissible use of vaping within hospital grounds as a smoking cessation tool [42]. Consistency in policy was also echoed by Farrimond and Abraham (2018), who emphasised wholescale changes in practice [35]. We would also recommend considering and exploring the de-implementation of practices that may be counterintuitive to promoting enrolment in e-cigarettes as a smoking cessation tool.

Our findings add to the ongoing debate surrounding the controversy of e-cigarettes as a smoking cessation tool in pregnancy. This study and a related study conducted by our team (Lutman-White et al., under review) provide useful insights that highlight significant challenges that need to be overcome before e-cigarettes are to be part of stop smoking services. Whilst our findings are positive, we must not ignore the ethical concerns that were raised and need to be alleviated through ongoing training for healthcare professionals. There is accumulating qualitative evidence repeatedly highlighting the hesitancy experienced by healthcare professionals when advising and providing e-cigarettes as part of their practice. In theory the use of e-cigarettes as a harm-reduction tool is understood, but implementation in practice, where there are strong public perceptions (both negative and positive) and mixed evidence around safety, efficacy and benefits, means more support and attention needs to be given to the difficulties healthcare professionals can experience. This needs to be a priority before the implementation of e-cigarettes in stop smoking services is considered.

## 5. Conclusions

Good enrolment in the e-cigarette pilots was demonstrated, with notable quit rates among pregnant women. We present useful insights into barriers to pilot implementation, which included inconsistency in the perceptions of e-cigarettes among relevant services and resource availability. The exploration of intervention set-up and provision demonstrates the extensive work that was undertaken to engage key stakeholders and opinion leaders at different stages, which ultimately facilitated implementation. A clear and in-depth understanding of the needs and resources of those served by the stop smoking in pregnancy service was central to pilot inception, and intervention providers considered that the pilot engaged pregnant women who the team would not usually successfully engage. Intervention providers sought the future rollout of e-cigarettes to be delivered in line with the standard treatment offer.

## Figures and Tables

**Figure 1 ijerph-21-00291-f001:**
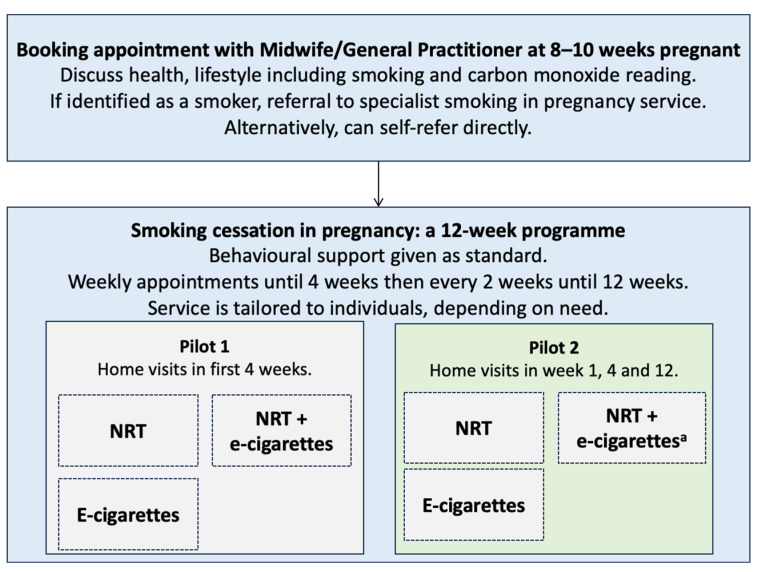
Flowchart to illustrate the pathway for pregnant women at the two sites. ^a^ Additional home visit at 8 weeks.

**Figure 2 ijerph-21-00291-f002:**
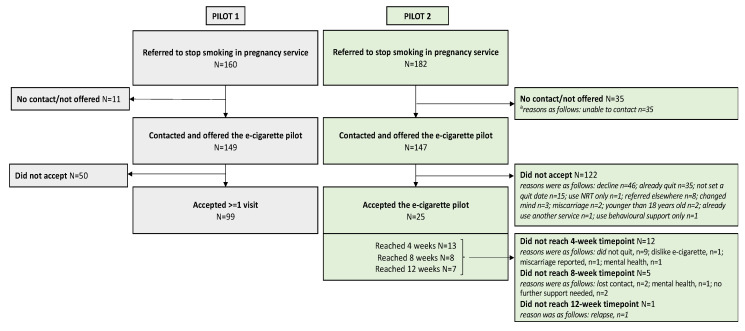
Flowcharts for the two e-cigarette pilots.

## Data Availability

All data generated or analysed during this study are included in this published article.

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
