# Peer review of "Implementing E-Cigarettes as an Alternate Smoking Cessation Tool during Pregnancy: A Process Evaluation at Two UK Sites"

_ijerph, 2024, doi:10.3390/ijerph21030291_

Round 1
Reviewer 1 Report
Comments and Suggestions for Authors
The introduction provides a concise and clear review of the literature on stop smoking services in pregnancy and increasing e-cigarette use.
The introduction could be strengthened by discussing how evaluating implementation could impact future practice or studies. That is, explaining why it would be helpful to evaluate implementation of the pilot studies while there is not yet strong evidence for e-cigarettes for quitting smoking during pregnancy.
Methods
The language used regarding “uptake of the pilot” could be clarified throughout the paper. This seems to be pregnant women’s acceptance of the e-cigarette intervention.
Setting
What implementation strategies were employed in the implementation of each pilot?
Limitations
The authors should note that only 7 individuals were interviewed.
Author Response
Reviewer #1
The introduction provides a concise and clear review of the literature on stop smoking services in pregnancy and increasing e-cigarette use.
The introduction could be strengthened by discussing how evaluating implementation could impact future practice or studies. That is, explaining why it would be helpful to evaluate implementation of the pilot studies while there is not yet strong evidence for e-cigarettes for quitting smoking during pregnancy.
We have added in a sentence to the introduction:
“Evaluating the implementation of pilot interventions is important to help guide future practice.”
Methods
The language used regarding “uptake of the pilot” could be clarified throughout the paper. This seems to be pregnant women’s acceptance of the e-cigarette intervention.
We have adjusted our language to use ‘enrolment’ instead of ‘uptake’ throughout.
Setting
What implementation strategies were employed in the implementation of each pilot?
We have added a sentence that the two pilots were implemented with behavioural support alongside the intervention:
“An implementation strategy employed by both pilots was the provision of behavioural support alongside the e-cigarette intervention.”
Limitations
The authors should note that only 7 individuals were interviewed.
We have added in 7 to the limitations:
“Interviews were conducted with seven pilot providers…” and “However, a strength was that the professionals involved in setup through to delivery were interviewed.”
Reviewer 2 Report
Comments and Suggestions for Authors
Dear authors, thank you for your efforts on this manuscript. I understand that your work does not only include that which you describe in this manuscript, but also i. a. extensive discussions with stakeholders in the UK Public Health System, and a review of the current evidence-base for using e-cigarettes as a means of supporting quitting smoking in pregnant women. That you have been able to do what you did on a topic as controversial as this one is commendable. I feel that if you revise the manuscript, it will be a worthy addition to scientific literature.
My main comments to your paper are that I (1) strongly suggest adding a paragraph on the controversial nature of this topic - in an effort to reach healthcare professionals who support pregnant women and who may be biased against the use of e-cigarettes as a means to quit smoking; (2) miss some information in the Methods sections; (3) think the Results section can be improved. Please find my specific comments in the document attached.

Author Response
Reviewer #2
Dear authors, thank you for your efforts on this manuscript. I understand that your work does not only include that which you describe in this manuscript, but also i. a. extensive discussions with stakeholders in the UK Public Health System, and a review of the current evidence-base for using e-cigarettes as a means of supporting quitting smoking in pregnant women. That you have been able to do what you did on a topic as controversial as this one is commendable. I feel that if you revise the manuscript, it will be a worthy addition to scientific literature.
Thank you for sharing your positive perspective of our work.
My main comments to your paper are that I
(1) strongly suggest adding a paragraph on the controversial nature of this topic - in an effort to reach healthcare professionals who support pregnant women and who may be biased against the use of e-cigarettes as a means to quit smoking;
(2) miss some information in the Methods sections;
(3) think the Results section can be improved.
Please find my specific comments in the document attached.
Please find our responses to each comment below.
General
The topic of the manuscript is controversial: offering e-cigarettes or NRT to pregnant women instead of regular cigarettes sounds like replacing one evil with another – even though expert opinion may prefer e-cigarettes / NRT over regular cigarettes. I feel the manuscript would benefit if the authors take some time to reflect on how controversial the topic can feel to readers / health care professionals (GPs, gynaecologists, nurses, ...) less familiar with interventions around smoking cessation, but who could play a role in helping pregnant women quit smoking. This could provide some perspective to the controversy, and have relevant healthcare professionals be more open to think of these kinds of interventions when they engage with smoking pregnant women in their practice – hopefully eventually resulting in more women quitting smoking during pregnancy.
Thank you for your comment. Whilst we agree that this is a controversial, we have attempted to address this comment in our discussion. We recognise that healthcare professionals particularly those involved in stop smoking services have difficulties in knowing what is the “right or wrong” answer when it comes to e-cigarettes, so we have added some reflections around this and that addressing these should be a priority before e-cigarettes are implemented more widely.
Our findings add to the ongoing debate surrounding the controversy of e-cigarettes as a smoking cessation tool in pregnancy. This study and a related study conducted by our team (Lutman-White et al, under review), provide useful insights that highlight significant challenges that need to be overcome before e-cigarettes are to be part of stop smoking services. Whilst our findings are positive, we must not ignore the ethical concerns that were raised, and need alleviating through ongoing training for healthcare professionals. There is accumulating qualitative evidence repeatedly highlighting the hesitancy experienced by healthcare professionals when advising and providing e-cigarettes as part of their practice. In theory the use of e-cigarettes as harm reduction tool is understood, but implementation in practice where there are strong public perceptions (both negative and positive), mixed evidence around safety, efficacy, and benefits means more support and attention needs to be given to the difficulties healthcare professionals can experience. This needs to be a priority before implementation of e-cigarettes in stop smoking services for pregnancy and beyond is considered.
Abstract
- Line 15: advise to not use the abbreviation ‘NRT’, but to write ‘nicotine replacement therapy
We have amended this in response.
Introduction
- International Journal of Environmental Research and Public Health has a broad scope, and not all readers may be familiar with smoking cessation strategies. Therefore, I suggest including a short sentence on what NRT entails, and a short explanation on nicotine metabolism during pregnancy.
We have added in two sentences that describe NRT and nicotine metabolism during pregnancy:
“NRT delivers a controlled amount of nicotine without the chemicals found in cigarettes and often used to reduce the urge to smoke. In pregnancy, women metabolise nicotine at a faster rate and so this must be factored in to smoking cessation support.”
- Line 48: what is ‘combination NRT’? Is there a difference between that and ‘just’ NRT?
We have added in clarification:
“…combination NRT (two forms of NRT together) in…”
Methods
- Line 102: how long does a disposable e-cigarette last?
We have added in detail around disposable e-cigarette lifespan, as per the following:
“…with a single use, disposable e-cigarette (provides 320 x 2 sec puffs; roughly equates to 30-35 cigarettes; approx. 2-4 days), alongside…”
- Line 103: if you offer both an e-cigarette and NRT to pregnant women, don’t they then get a double dose of nicotine?
We have added clarity in the earlier paragraph of the Context section:
“Both pilots aligned with recommended practice where pregnant women should be offered two forms of nicotine replacement therapies during a structured quit attempt (typically patches to match nicotine levels the body is used to and another form to cope with the physical cravings, in this instance through e-cigarettes).”
- Line 103: NRT is spelled out as a whole, but that is unnecessary as the abbreviation was introduced in line 45.
We have deleted ‘nicotine replacement therapy’ in response.
- Line 105: what are CO readings? Can you deduce smoking habits from that? I suggest adding a short explanation.
We have added clarification here:
“…verified by CO readings (to deduce smoking habits objectively by detecting carbon monoxide in exhaled breath).”
- Line 106 + 110: what did the home visits look like? If behavioural intervention was part of those visits, what did that look like in general? Could you include a few lines on that – or refer to a paper / guideline with a description?
We have added information about home visits:
Pilot 1- “During these visits behavioural support is offered, to include information on how to deal with cravings, distraction techniques, what happens to your body when you quit, as well as relapse prevention strategies…”
Pilot 2- “The content and intention of the home visits and telephone/text mirrored the support described in Pilot 1.”
- Line 112: similar comment as the previous one: what does phone/text support look like? Could you either include a short line on that, or refer to a paper / guideline with a description?
We have added some information here:
“Pregnant women also had access to follow up calls and texts with the advisor as and when needed in between appointments. This form of ad-hoc support acted as an important motivator and trust building.”
- Line 113: it says that one advisor delivered pilot 2. Who delivered pilot 1?
We have clarified:
Pilot 1- “…and was delivered by multiple advisors in the Health in Pregnancy Team.”
- Participant selection: I am not entirely clear on how the process went for pregnant women. Did they also consent in Qualtrics? Were they aware that they were taking part in a pilot? I suggest clarifying the paragraph, potentially by splitting it in two: first part on pregnant women who participated and second part on intervention providers. Also, if possible, I suggest adding some demographics about the pregnant women.
We have clarified and also split this section into text around the pilot and text around the qualitative interviews.
“Information about the pilots was offered at time of booking and therefore aware of the e-cigarette offer before their first appointment, during which written consent to participate in the pilot was obtained.”
- Data collection: how was determined who really quit smoking during the intervention? I see something in Line 419 of the Discussion; I suggest adding information about the procedure to the Methods section.
We have clarified quits were CO verified on most occasions:
“Given that home visits were undertaken in these pilots, the majority of quits were CO verified (otherwise self-reported).”
- Data collection: what were the interview questions? Where these structured / unstructured / semi-structured interviews? It would be helpful to spend a few lines in CFIR –> there is some information in Data Analysis. I suggest moving that upwards, either to Data Collection, or, even better, to have a short paragraph dedicated to CFIR. I feel the information does not belong in Data Analysis.
In response, we have moved some text up from the data analysis section to the data collection section. We have also clarified ‘semi-structured’ in this section too.
- A potentially difficult question: the CFIR framework you based your interviews on was updated in 2022. In what ways would your analysis be different if you had been able to use the updated framework compared to the one from 2009?
This is a difficult question – the new CFIR framework in 2022 is much improved, with some similarities of terminology in the domains and constructs to the original version used in our research. However, when comparing the new framework against the old, the introduction of new sub-constructs under the original construct external policies and incentives, would have been able to capture findings related to financing and the role it plays in roll out e-cigarettes, as we all as external pressures sub-construct which could have captured the role of public perception and how that influences implementation and pre-implementation strategy, as during our pilot there was a lot of focus within central government around e-cigarettes in particular amongst young people. This is something our future work could definitely explore further.
Results
- I find it confusing that the results for pilots 1 and 2 are mixed, e.g. ‘In total 124 women accessed at least one visit from either pilot’. I suggest first giving the information for pilot 1, and then for pilot 2. Then, if desired, the combined results could also be reported.
We have amended so that the numbers are split by pilot:
“In relation to pilot 1 and pilot 2 sites, 160 and 182 women, respectively, were referred to the stop smoking in pregnancy service in the specified timespans. In pilot 1, 99 women accessed at least one visit from either pilot. In pilot 2, 25 accepted the pilot offer with the most common reason for not accessing the pilot relating to already quitting (N=35).”
- Are there data whether there are (demographical / SES / ...) differences between the women who accepted the service offer or not, and who eventually quit or not (although N may be too small for the latter)?
No, this data is not available to be reported on.
- Figure 2: for pilot 2 there is information as to why 35 eligible women were not contacted (‘unable to contact’). Is it possible to add this information for the 11 eligible, uncontacted women in pilot 1? If not, why was the information not collected?
This data was not collected in pilot 1 so we have been unable to report on it.
- Figure 2: similar as previous comment: what are the reasons why women in pilot 1 did not accept the offer, and if this information is not available, why not?
As above, this data was not collected in pilot 1 so we have been unable to report on it.
- Figure 2: similar as previous comment: for pilot 2 there is quite some information on how many women were reached at different weeks. Why is this information not available for pilot 1. If I understand correctly from the Methods section, there should be information about this.
This data was not collected in pilot 1 so we have been unable to report on it. In the methods we refer to “In pilot 1, engagement data related to women engaging with at least one visit. In pilot 2, data were collected about engagement at 4, 8 and 12 weeks.” where we feel this clarifies the lack of detailed information about engagement beyond the initial visit.
- Line 180: it says that pilot 1 did not differentiate the proportion who used e-cigarettes compared to NRT. I am not sure what you are saying here. Do you mean that as pilot 1 did not differentiate, you do not know whether the quits can be attributed to the NRT or the e- cigarettes? Could you clarify? And how is this for pilot 2?
We have added clarity around this for both Pilot 1 and Pilot 2:
Pilot 1- “Pilot 1 did not differentiate the proportion who used e-cigarettes alone compared to e-cigarettes plus NRT.”
Pilot 2- “In pilot 2, most women selected e-cigarettes exclusively.”
- Box 1: do I correctly understand that the box represents the CFIR domains and associated constructs in general, and with the ‘a’ you have indicated which domains and constructs you identified in the interviews? If not, could you please clarify? Furthermore, in line with my comments on the Methods section, it would be helpful to already introduce this box in the Methods section, in a paragraph dedicated to CFIR.
Yes, that is correct. We feel our modifications in response to the earlier comment now helpfully enable the reader to understand the domains of the CFIR. Then, here in the results, the reader is shown the domains and constructs explicitly whilst referring to which relate to the qualitative work done.
- Qualitative findings: small, general comment: some of the quotes from the interviewees are italicised, others are not.
We have checked the manuscript and cannot see any non-italicised quotes; however, during the proofing stage we will ensure this formatting is consistent throughout.
- Lines 210 + 226: very relevant observation, that e-cigarettes use is not without risk, and that e- cigarettes are not always perceived as quitting tool, but as something for longer-term use. This ties in with my general comment about the controversial side of an intervention such as this; healthcare providers (and readers) unfamiliar with the particulars may be biased against advising it to pregnant women – although in the end it may help women quit. This is also linked to the first part of “Inner setting”, about perceptions and acceptance of e-cigarettes, and to line 344 stating that “e-cigarette use could evoke strong opinions [in colleagues]”. This is worth mentioning in the discussion -> you do this to a certain extent, but you can extend this.
We have checked the discussion and, to reflect the data and how we have interpreted it, we have made some changes to add a brief reference to the sensitive nature of e-cigarettes in this section:
“The importance of alleviating safety concerns, associated risks and sensitive views with using e-cigarettes among healthcare professionals has been identified as crucial within and beyond our findings.”
- Line 245: it includes a comment about health inequalities in the area targeted by pilot 2. It is useful to provide this kind of information in the methods section, so some characteristics of (the population in) the targeted area.
We have added the following sentence into the methods to reflect on pilot delivery in two regions of interest:
“The pilots targeted areas with highest smoking prevalence, socioeconomic deprivation, mental health deprivation, and health inequalities.”
- Line 377: it states that prior to the pilots there had been systematic review of the evidence- base and key stakeholders had been involved. I feel this is information that could be added to the Methods section, or a reference to a paper if this has been published elsewhere.
We intend for the full citation to be included in the published version of this paper. At the point of submission, though, we were advised to ‘blind’ our author details as much as possible and, as we were involved in writing that systematic review, we left the “(authors citation blinded)” phase in there in its place. We will ensure a full reference to the paper is included in the proof stage.
Discussion
- Line 441: “the team felt they had engaged with people they would not usually engage with.” This is an important finding. It may be helpful to include something about hard-to-reach populations somewhere in the Introduction or description of the participant sample. An additional question about this: were you aiming specifically to reach hard-to-reach target groups? If so, it is also worth mentioning earlier in the manuscript.
This was a positive unintended consequence of this research and so we have not added any information about targeting of hard-to-reach groups in the methods. It was certainly a positive outcome in which advisors felt they engaged with women they would not ordinarily engage with, so we have highlighted this as a strength in response to your next comment.
- Line 467: as a limitation it is mentioned that the pilots targeted specific areas with higher smoking rates, lower SES and health inequalities, so results may not be generalisable. My first comment on this is that it is helpful– as I mentioned somewhere earlier – to include this descriptive information somewhere in the Methods section. A second comment is that this is not only a limitation, but also a strength: you have shown that you were able to reach this hard-to-reach group. This group may be in most need of support, so although the results may not be generalisable to the general population, they may be generalisable to other low SES / health inequal areas in the UK who may be more in need than other areas, and that is very valuable indeed.
Thank you for making us reflect on this. In response, we have rephrased to:
“The pilots managed to engage specific areas of their respective counties (regions) with higher smoking rates, socioeconomic deprivation and health inequalities and therefore the results may be generalised to pregnant women sharing similar demographics.”
- Line 475: do I understand correctly that you found the same quit rate after 12 weeks as the NHS statistics after 4, ánd which – in contrast to NHS statistics – you verified much more consistently with CO measurements? If so, this is an achievement you can be proud of. If I read this wrong, please clarify
We have rephrased to be clearer. Yes, the pilot 2 quit rate is comparable to NHS statistics at 4 weeks but the added strength here is that these quits were CO verified.
“Just over half of pregnant women enrolled in pilot 2 had quit smoking by 12 weeks (CO verified) whilst a lower quit rate (32.3%) was recorded in pilot 1. Quit rate for pilot 2 (52%) is similar to that reported for pregnant women by the NHS in their statistics on NHS Stop Smoking Services in England (53.6%); although, the NHS figure relates to quits at the 4-week follow up and only 13.6% of those quits were CO verified (NHS, 2021). This is a key finding and a strength of the pilot 2 work in particular, recording verified quits at 12 weeks.”
- Line 476: if I understood the previous bit accurately, and you found a pretty OK quit rate compared with NHS, then there is no need to start the next sentence with “However”. This makes it feel as though your result is not that good compared to other existing literature (it feels like you are going to say “However, authors x and y found z, which is way better than our outcome”), where in fact you are saying that from your review you found that “the efficacy of e-cigarettes as a tool for stopping smoking during pregnancy remains unclear”. It would be a shame to talk your findings down, so I suggest removing “However”. Instead, you could say something that comparing to the general literature is complex because “the efficacy of e- cigarettes as a tool for stopping smoking during pregnancy remains unclear.”
We have amended this sentence to remove the word ‘However’.
- In the study you documented several barriers s to pregnant women engaging in the e-cigarette pilot, including the home visits, the influence of family members and previous negative experiences with professionals. It would be good to include some ideas for future work: what study and / or what clinical practices could be tried for addressing these barriers? You do mention some alternatives for the home visits, but it would be nice to speculate on the other barriers, too.
We have added reference to the encouragement of including family members in to service alongside pregnant women. Amended sentence here:
“Changes to improve burden of delivery could include reducing the number of visits, encouraging family members to engage in the service in parallel, and exploring other options for carbon monoxide monitoring (such as midwife or self-testing).”
Reviewer 3 Report
Comments and Suggestions for Authors
From the formulation of the study aim, it can be concluded that this study aimed to evaluate the implementation of two UK-based pilots and explore the views of stakeholders on implementation and scalability. Information about pregnant women’s experiences with using e-cigarettes during pregnancy is reported elsewhere.
However, the Materials and Methods section predominantly focuses on describing the pilot study involving pregnant women and lacks details on how the views of stakeholders were explored. The Results section reveals that there were 7 individuals involved. It is crucial to provide a detailed description of this aspect in the Materials and Methods section, as it constitutes the core of the present manuscript. Therefore, the subsections titled "Participant Selection: Sampling," "Setting," and "Data Collection" should largely refer to those stakeholders.
Regarding the "Results" section, it aligns with the study aim. However, in the Discussion section, greater emphasis should be placed on the stakeholders and their views regarding the implementation and scalability of the study.
Author Response
Reviewer #3
From the formulation of the study aim, it can be concluded that this study aimed to evaluate the implementation of two UK-based pilots and explore the views of stakeholders on implementation and scalability. Information about pregnant women’s experiences with using e-cigarettes during pregnancy is reported elsewhere.
However, the Materials and Methods section predominantly focuses on describing the pilot study involving pregnant women and lacks details on how the views of stakeholders were explored. The Results section reveals that there were 7 individuals involved. It is crucial to provide a detailed description of this aspect in the Materials and Methods section, as it constitutes the core of the present manuscript. Therefore, the subsections titled "Participant Selection: Sampling," "Setting," and "Data Collection" should largely refer to those stakeholders.
Regarding the "Results" section, it aligns with the study aim. However, in the Discussion section, greater emphasis should be placed on the stakeholders and their views regarding the implementation and scalability of the stud
We are presenting a mixed-methods process evaluation of the two pilots and have attempted to offer a balanced insight into the quantitative and qualitative findings throughout. We politely disagree that the discussion does not have a strong emphasis on the qualitative data presented in the results, and feel there is a good balance there. However, we agree there was a need to increase the visibility of stakeholder views and so in response we have made a few changes:
(1) The methods sections are now structured with text on the pilot (quant) and interviews (qual) separately i.e., sampling, setting, data collection, and data analysis.
(2) We have added more detail on the data collection section for the stakeholder interviews around the CFIR framework.
Round 2
Reviewer 3 Report
Comments and Suggestions for Authors
Well done!